# Triangle-Mesh-Rasterization-Projection (TMRP): An Algorithm to Project a Point Cloud onto a Consistent, Dense and Accurate 2D Raster Image

**DOI:** 10.3390/s23167030

**Published:** 2023-08-08

**Authors:** Christina Junger, Benjamin Buch, Gunther Notni

**Affiliations:** 1Group for Quality Assurance and Industrial Image Processing, Technische Universität Ilmenau, 98693 Ilmenau, Germanygunther.notni@tu-ilmenau.de (G.N.); 2Fraunhofer Institute for Applied Optics and Precision Engineering IOF Jena, 07745 Jena, Germany

**Keywords:** computer vision, single-shot projection, upsampling, interpolation, registration, data fusion, depth completion, depth image enhancement, sparse data, transparent object

## Abstract

The projection of a point cloud onto a 2D camera image is relevant in the case of various image analysis and enhancement tasks, e.g., (i) in multimodal image processing for data fusion, (ii) in robotic applications and in scene analysis, and (iii) for deep neural networks to generate real datasets with ground truth. The challenges of the current single-shot projection methods, such as simple state-of-the-art projection, conventional, polygon, and deep learning-based upsampling methods or closed source SDK functions of low-cost depth cameras, have been identified. We developed a new way to project point clouds onto a dense, accurate 2D raster image, called Triangle-Mesh-Rasterization-Projection (TMRP). The only gaps that the 2D image still contains with our method are valid gaps that result from the physical limits of the capturing cameras. Dense accuracy is achieved by simultaneously using the 2D neighborhood information (rx,ry) of the 3D coordinates in addition to the points P(X,Y,V). In this way, a fast triangulation interpolation can be performed. The interpolation weights are determined using sub-triangles. Compared to single-shot methods, our algorithm is able to solve the following challenges. This means that: (1) no false gaps or false neighborhoods are generated, (2) the density is XYZ independent, and (3) ambiguities are eliminated. Our TMRP method is also open source, freely available on GitHub, and can be applied to almost any sensor or modality. We also demonstrate the usefulness of our method with four use cases by using the KITTI-2012 dataset or sensors with different modalities. Our goal is to improve recognition tasks and processing optimization in the perception of transparent objects for robotic manufacturing processes.

## 1. Introduction

### 1.1. Challenges in Single-Shot Projection Methods

In various tasks of image analysis and enhancement (Section 1.2), points P(X,Y,V) (the value V stands for any modality, e.g., for depth or disparity) must be projected onto a 2D raster image. Unfortunately, the state-of-the-art (SOTA) single-shot projection methods (Section 2) still have challenges that need to be solved, particularly when a low-resolution point cloud is to be projected onto a high-resolution 2D raster image. Figure 1 shows these challenges (A–D). Since the 3D coordinates X and Y are usually not integers, in the 2D image, the *V* value must be distributed among the surrounding four 2D pixels (Figure 1). Real gaps that exist due to physical limitations of the camera technology [1,2] used may not be fully considered (A). If neighboring 3D X/Y coordinates are further than one pixel apart, then false gaps between these pixels in the 2D image are created (B). With common single-shot projection methods, error (B) only occurs when using different hardware due to different resolution and viewing angle (see Section 6.1.4). If neighboring 3D X/Y coordinates are less than one pixel apart, then false neighborhoods between these pixels in the 2D image are created (C). This challenge occurs in SOTA projection methods as a fattening problem near depth discontinuities [3,4,5,6]. Additionally, the foreground and background are not cleanly separated (D).

To increase the processing stability of the various image analyses and enhancements (Section 1.2) in the future, we have developed a new way to project points onto a dense, accurate 2D raster image, called Triangle-Mesh-Rasterization-Projection (TMRP). Our method solves the above challenges (A–D). Figure 2 shows a direct comparison of the differences between the simple SOTA projection and our TMRP. Section 2 describes and compares other SOTA approaches.

### 1.2. Use Cases of Single-Shot Projection Methods

The projection of points onto a 2D raster image is relevant for various image analysis and enhancement tasks, especially when low-resolution point clouds need to be merged into a high-resolution image (or point cloud). Figure 3 shows generalized possible measurement arrangements: (left) in multimodal image processing, (mid) in robotic applications or in scene analysis, and (right) for deep neural networks.

#### 1.2.1. General: Data Fusion of Multimodal Image Processing

The relevance of multimodal systems is also increasing nowadays. This enables a more comprehensive understanding of the real environment [8], e.g., for autonomous navigation [9], medical applications [10,11], for quality control [12], for interactive robot teaching [13], for safe robot–human cooperation [14], or in scene analysis [15,16,17]. The projection of transformed points onto a 2D raster image is often used in multimodal image systems for data fusion resp. registration [9,18,19] (Figure 3, left). The challenges that arise in image data fusion [20] are: non-commensurability, different resolutions (challenge (B)), number of dimensions, noise, missing data, and conflicting, contradicting, or inconsistent data (challenge (D)).

#### 1.2.2. Robotics Applications or Scene Analysis—The Need for Dense, Accurate Depth Maps or Point Clouds

Depth sensors are widely used in robotics applications in terms of robot teaching [13], navigation, grasp planning, object recognition, and perception [14,15,21,22]. Sensors such as Orbbec3D Astra Pro or Azure Kinect provide color point clouds at 30 fps, which is of particular interest for human–robot collaboration [22,23]. The fusion of camera data and modern 3D Light Detection and Ranging (LiDAR) data has many applications, including autonomous and safe control of mobile robots [18], object detection [8], and simultaneous localization and mapping (SLAM) [24]. Depth sensors are also widely used in scene analysis and human–computer interaction, e.g., object detection [8,17,25], (semantic) segmentation, [15,16,17] or markerless motion capture applications [26].

Depending on the application, the acquired point clouds are processed in 2D [8,15,16,18,27] or in 3D [13,14,26]. A projection of the (low-resolution) points onto a (high-resolution) 2D raster image or point cloud is required (Figure 3, mid). Different current single-shot projection methods can be used for this purpose (Section 2). For consumer depth/RGB-D sensors, the projection of point clouds is done via the software development kit (SDK) provided by the manufacturer, whereby a closed source function is available. However, current single-shot projection methods cannot fully solve the challenges (see Section 6).

#### 1.2.3. Deep Neural Network—The Need for a Large Amount of Dense, Accurate Correspondence Data

Deep learning networks have gained enormous importance in recent years, e.g., in monocular [15,28] and deep stereo [29,30,31,32,33,34,35,36] frameworks. The main reasons for the limitations of deep frameworks—highly data-driven methods—are: (i) challenging handling of sparse input data [37], (ii) the lack of suitable datasets for their own application (e.g., existing outdoor driving [27,38] instead of industrial scenarios, or differences in camera arrangement (parallel instead of convergent) [39]), and (iii) the high effort required to create real datasets with dense, accurate correspondence, also called ground truth (Figure 3, right) [28,34,36]. The density of the correspondence depends on the reference system and method used (cf. KITTI-2012 [27] and Booster, the first stereo benchmark dataset for several specular and transparent surfaces [36]). Depending on the measurement system and setup used, the single-shot projection method also has an impact on the density of the generated data (cf. experiments #1–#4 in Section 6).

In the field of deep stereo, for example, the generation of real training data with dense ground truth disparities is very complex (costly and time-consuming), especially for visually uncooperative objects in the visible spectral range [1,36,40,41], e.g., specular, non-reflective, or non-textured surfaces. Here, the painting of uncooperative objects is SOTA [36,40,41]. However, in collaboration with the Fraunhofer Institute IOF, we have developed a new method without object painting based on a novel 3D thermal infrared sensing technique by Landmann et al. [42]. In experiment #4 in Section 6, a sample frame from this dataset is used. Here, the advantage of our TMRP algorithm can be clearly seen. Due to reasons (i)–(iii) above, synthetic datasets [43,44,45] are mostly used, which provide numerous and dense datasets with less effort. Another possibility is the use of semi-synthetic datasets [34]. Despite the fact that synthetic data offer enormous possibilities, the big challenge is to close (performance) differences between real and synthetic. There are several differences, such as differences in distribution [34,35], labeling, and photorealism [46].

### 1.3. The Main Contributions of Our Paper

The main contributions of our paper are:We propose a novel algorithm, called Triangle-Mesh-Rasterization-Projection (TMRP), that projects points (single-shot) onto a camera target sensor, producing dense, accurate 2D raster images (Figure 2). Our TMRP (v1.0) software is available at http://github.com/QBV-tu-ilmenau/Triangle-Mesh-Rasterization-Projection (accessed on 1 July 2023).To fully understand the components of TMRP (Algorithm 1), we have written down the process and mathematics as pseudocode (see Appendix B). In addition, there is Appendix A for the mathematical background of some experiments (Section 6).Other single-shot projection methods are discussed, evaluated, and compared.We believe that our TMRP method will be highly useful in image analysis and enhancement (Section 1.2). To show the potential of TMRP, we also present several use cases (cf. measurement setup in Figure 3) using qualitative and quantitative experiments (Section 6). We define performance in terms of computation time, memory consumption, accuracy, and density.

## 2. Related Work

There are projection methods based on single-shot and multiple-shots per camera/sensor, including RGB-D Simultaneous Localization and Mapping (SLAM) [47,48], SDF field [49], and Iterative Closest Point (ICP) technique [50]; see also Point Cloud Registrations in [51].

We focus on single-shot projection methods. The representatives are listed below:(i)Simple SOTA projection (Algorithm 1).(ii)Upsampling methods—see also depth image enhancement [5,52]:Conventional methods: The most commonly used interpolation approaches are nearest-neighbor, bilinear [7], bicubic, and Lanczos [53] schemes.Pros: Simple and present a low computational overhead.*Con:* Achieved visual quality is low due to strong smoothing of edges. Cause: linear interpolation filters have a smooth surface from the start [3,6].Further methods are: Joint bilateral upsampling (JBU) [6], which achieves better results. This method works through a parameter self-adaptive framework [19], Gaussian process regression (GPR), or covariance matrix-based method [18].Polygon-based method—a research area with very little scientific literature [7]. The Delaunay triangles and nearest neighbor (DT_nea_) [7,8] based on LiDAR data alone, i.e., color/texture information from the camera, is not used (see Section 5).Pros: Inference from real measured data; the interpolation of all points is done regardless of the distance between the points of a triangle. This approach uses only data from LiDAR. The extra monocular camera is only considered for calibration and visualization purposes.Cons: Complexity of the algorithm is very high; dependence of results on data quality; noise or outliers in the input data can be amplified or generated; too high upsampling rates can distort the result and make it inaccurate.Deep learning-based upsampling: Also known as depth completion (where a distinction is made between methods for non-guided depth upsampling and image guided depth completion [37]). Here, highly non-linear kernels are used to provide better upsampling performance for complex scenarios [3]. Other representatives: [5,30,32,54,55].Pros: Better object delimitation, less noise, high overall accuracy.Cons: Difficult generalization to unseen data; boundary bleeding; training dataset with ground truth is required.(iii)Closed sourced SDK functions from (consumer) depth sensor manufacturers.

In all the above methods, the resulting projected 2D raster image has gaps or false neighbors. The error rate depends on the projection method used. On the one hand, true gaps caused by physical limitations of the camera technology (A) are not considered. On the other hand, false gaps (B) or false neighborhoods (C) are created. In addition, the density is XYZ-dependent (resp. XYV), with the exception of the polygon-based upsampling method. Moreover, some methods cannot resolve ambiguities (D). This includes the *simple SOTA projection* (Algorithm 1). Challenge (B) occurs for cameras with different resolutions. For example, the spatial resolution of common depth technologies is much lower than that of cameras. To match the resolution of these two sensors and thus obtain a dense depth map, the false gaps in the depth map need to be interpolated [7,18]. The deviation of these estimated values from the corresponding actual values depends on the projection method used. *Joint bilateral upsampling (JBU)* [6] tries to increase density by down- and upsampling. Challenges (A) and (C) occur even more frequently than with simple SOTA projection. *Joint bilateral propagation upsampling (JBPU)* is a way to accelerate and densify unstructured multi-view stereo, which builds on a JBU and depth propagation strategy [56]. Compared to JBU [6] and bilateral guided upsampling (BGU) [57], denser depth maps are achieved here. However, the aforementioned challenges exist here as well. *Deep learning methods* are another alternative to increase density, e.g., [28,32,54], which aim to solve this problem based on a large amount of data. However, the challenges described above also arise with this method [36,39,58]. Another example is the CNN SparseConvNet developed by Uhrig et al. [37] that can be used to complete depth from sparse laser scan data. For more details on how to handle sparse input data resp. invalid pixels and their limitations, see Uhrig et al. [37]. The deep learning methods are the least accurate compared to SOTA projection and JBU. Point clouds from (low-cost) depth sensors, such as Basler blaze-101, Orbbec3D Astra Pro, or Azure Kinect, are usually projected onto a depth image using *closed sourced SDK functions* provided by the manufacturer. The initial problems are solved in the modern RGB-D sensors [21,59]. However, there are still challenges that have not been solved [60], such as the incomplete consideration of gaps in projection, challenge (A). This method is more accurate compared to the simple SOTA projection.

## 3. Processing Pipeline When Using TMRP

The cameras or sensors used are calibrated, e.g., according to Zhang [61] and synchronized beforehand so that the data from the source camera/sensor match the corresponding image from the target camera/sensor. The fusion of different camera/sensor data requires the extrinsic calibration of the cameras/sensors [18], which are needed for the first step (the coordinate transformation). Depending on the camera technologies used, calibration targets with multimodal illumination [14] can be used to obtain more robust calibration results.

The input of our TMRP is a PLY point cloud with transformed points P(X,Y,V,rx,ry). Figure 4 shows the necessary steps to generate transformed points using a measurement camera system based on a camera and a depth sensor.

Coordinate transformation: The source point cloud $P_{\mathrm{source}}\left(X,Y,V\right)$ is transformed into the coordinate system of the target camera using the calibration parameter and then into the image plane. (In the Appendix A, the mathematical context is described for experiments #1 and #3. Another example is described for the KITTI dataset by Geiger et al. in [27,62].) In addition, the point cloud is extended by the raster information rx,ry of the source camera. The result is a planar set of points P(X,Y,V,rx,ry).Optional extension: E.g., conversion of depth to disparity (see experiments #3 and #4).TMRP: The points P(X,Y,V,rx,ry) can now be projected onto a dense accurate 2D raster image, called the *target image*, using TMRP (Section 4.2).Image to point cloud (optional): For applications based on 3D point clouds (Section 1.2), the dense, accurate (high-resolution) image must be converted to a point cloud P(X,Y,V) (Figure 4).

## 4. Explanation of TMRP and Simple SOTA Projection Algorithms

### 4.1. Overview

With almost all 3D measurement methods, 2D neighborhood information of the 3D coordinates can simultaneously be acquired. (The 2D raster rx and ry must contain integer values only.) If this information is available in the *x* and *y* directions as a property of the transformed points P(X,Y,V,rx,ry), it can be used to perform a dense, accurate interpolation between the 2D pixels that were adjacent in 3D. Figure 2 shows the comparison between naive SOTA projections and our TMRP. Our TMRP solves the SOTA challenges (A) to (D) described in Section 1.1. In addition, the density is XYZ independent. We present the individual components of our TMRP and SOTA projections in the following. Figure 2 shows the two methods and their resulting target images. Algorithm 1 defines some important parameters and provides an overview of our software (TMRP and SOTA proj.). Appendix B describes more detailed information, including the mathematical description, as pseudocode (Appendix B).
**Algorithm 1** User definitions and procedure for the Triangle-Mesh-Rasterization-Projection (TMRP) and state-of-the-art (SOTA) projection. For more details, see Appendix B.Input/output:point(X,Y,value)∨point(X,Y,value,rx,ry)←3Dpointofpointcloud(Appendix B, point)width←widthoftargetimage∈Nheight←heightoftargetimage∈Nraster_filter← max/min/nonetarget_image(target_value)[0:width−1,0:height−1]←targetimagePermissible value range:X,Y,value←x-andy-positionof3Dpoints∈Rrx,ry←rasterxandy-position∈Zmin←∈0∨1max←∈0∨1*(prefix)*: Raster (rx,ry)mustbeunique**procedure** Triangle-Mesh-Rasterization-Projection(point, width, height, raster_filter)    source_raster_image=createSourceRasterImage(point) → Appendix B, Figure 5a    vector_image=createVectorImageTMRP(source_raster_image, width, height) → Appendix B, Figure 6    vector_image=separationForegroundAndBackground(vector_image, raster_filter) → Appendix B, Figure 7    target_image=calculateTargetImage(vector_image, width, height) → Appendix B, Figure 2    **return** target_image**end procedure****procedure** SOTA-projection(point, width, height)    vector_image=createVectorImageSOTA(point, width, height) → Appendix B, Figure 5b    target_image=calculateTargetImage(vector_image, width, height) → Appendix B, Figure 2 (*top*)    **return** target_image**end procedure**

### 4.2. Triangle-Mesh-Rasterization-Projection

Our TMRP algorithm is divided into two parts (Figure 2, *bottom*): Part (I), creation of the vector image vector_image (Section 4.2.1), and part (II), the calculation of the target image target_image (Section 4.2.2).

#### 4.2.1. Part (I): Create Vector Image

This part consists of three sub-steps (Figure 2, *bottom*). For simplicity, we describe our algorithm using a transformed point cloud based on the measurement setup in Figure 4. ${\mathrm{cam}}_{2}$ is assumed to be the source camera, and ${\mathrm{cam}}_{1}$ is the target camera. If only one camera ${\mathrm{cam}}_{1}$ is to be used, it is considered as both source and target camera.

*(1)* 
*Create source raster image*


The source raster of ${\mathrm{cam}}_{2}$ is placed in the transformed point cloud using rx and ry (Figure 5a). To save memory, the 2D image is constrained to rx_range, ry_range. Appendix B describes the creation of source_raster_image in detail.

*(2)* 
*Create vector image*


This part describes the triangular interpolation between the 2D pixels that were adjacent in 3D. The goal is to find the 3D point that has the smallest distance to a grid point. The determined 3D points raw_pixel are appended with determined weight and interpolated value in vector_image(ix,iy) in lists. The mathematics for the generation of the vector_image are described in Appendix B.

Figure 6 shows the flow graphically for a better understanding of the process. The starting point is the previously created raster_image. For each raster point, a check is made to see if there are three or four 2D neighborhoods. If a neighborhood relationship exists, triangles are then drawn between the 3D points. A neighborhood of three results in one triangle, and a neighborhood of four results in four triangles (Figure 6). Bounding boxes are generated for each triangle. If a triangle is partially or completely out of the allowed range (outside of vector image), the inadmissible range (fx,tx∉[0,w−1] or fy,ty∉[0,h−1]) is ignored. However, the three points that span the triangle remain $\Delta (\mathrm{jp},t_{1},t_{2})$. For all grid points jp in the bounding box that lie in the triangle, the total triangle is decomposed into three sub-triangles: $\Delta (\mathrm{jp},t_{0},t_{1})$, $\Delta (\mathrm{jp},t_{1},t_{2})$, and $\Delta (\mathrm{jp},t_{2},t_{0})$. Afterwards, their areas are calculated according to Heron’s formula. In Appendix B, the total triangular area area_sum of the triangle $\Delta (t_{0},t_{1},t_{2})$ is calculated. However, due to numerical instabilities, in the software, the total area is calculated as the sum of the three partial triangle areas (${\mathrm{area}}_{0}$, ${\mathrm{area}}_{1}$, ${\mathrm{area}}_{2}$). After that, the weights ${\mathrm{weight}}_{k}$ of the triangle points are calculated for jp. The weights correspond to the area of the unconnected opposite triangle. Using the calculated weights and *V*-values of the 3D points, an interpolated value interpolated_value is calculated. The point $t_{k}$ that has the smallest distance to the point jp(jx,jy) (smallest distance means largest weight) is appended in the vector_image(ix,iy). A list in vector_image can consist of none/one/four entries (raw_points).

*(3)* 
*Separation of Foreground and Background*


The raster information (rx, ry) can also be used to cleanly separate foreground and background, as in challenge (D), Figure 2. This is especially useful for point clouds that have been transformed, as overlaps are very likely to occur. In marginal areas, however, this may already be the case without transformation. Depending on the application, a max/min/none filtering can be done. Figure 7 shows an example for V as disparity; here, a foreground selection is correct, as high disparity value is in the foreground and smaller disparity is in the background. For filtering, the maximum disparity value is determined as a reference value in the target pixel. Only values that are adjacent to this reference value in the raster are included in the target pixel. For more details, see Appendix B.

#### 4.2.2. Part (II): Calculate Target Image

Appendix B describes the calculation of the target_image based on the vector_image (from TMRP (Section 4.2.1) or SOTA projection (Section 4.3)), the target image width, and the target image height (Figure 8). To calculate the pixel value in target_image(i,j), value⊂vector_image(i,j) and weight⊂vector_image(i,j) from the respective list entries are used; see Equation (Equation 1). sum_weight_value and sum_weights of Equation (Equation 1) are described in Appendix B (2) and (3). As a by-product, we generate a confidence map that can be used for challenging applications that require masking treatment.
(1)$$\mathrm{target}\_\mathrm{image}\left(i,j\right)=\left\{\begin{array}{cc} \mathrm{NaN} & \mathrm{List}\mathrm{entry}=0\\ \mathrm{NaN} & \mathrm{sum}\_\mathrm{weight}=0\\ \frac{\mathrm{sum}\_\mathrm{weight}\_\mathrm{value}}{\mathrm{sum}\_\mathrm{weight}s} & \mathrm{else} \end{array}\right.$$

Our example target_image in Figure 8 is a pseudo-real disparity map (16-bit grayscale image with alpha channel). The alpha channel encodes the validity of each pixel. This channel can be used, for example, for convolutional neural networks (CNNs) to distinguish between observed and invalid input [37].

### 4.3. SOTA Projection

The SOTA projection is divided into two parts (Figure 2, *top*): Part (I), creating the vector image vector_image, and part (II), the calculation of the target image target_image. Part (I) is different from TMRP part (I) and is described below. Part (II) is described in Section 4.2.2.

Part (I): Create vector image SOTA—Appendix B describes the SOTA process without considering the raster information rx,ry (Figure 5b). Figure 1 shows the interpolation procedure and the resulting errors: (A) no consideration of true gaps if they are very small, (B) false gaps, (C) false neighborhoods, and (D) mixed foregrounds and backgrounds; see Section 1.1. In SOFT projection, a list in vector_image can consist of zero to *∞* entries (see Figure 8).

## 5. Comparison with Polygon-Based Method

The polygon-based method using Delaunay triangles and nearest neighbor (DT_nea_), used in [7], has similarities to our TMRP method. Both methods have the advantage that the density of the target image is not XYV-dependent. This is achieved because all points are interpolated independently of the distance between the points of a triangle.

The main differences with our TMRP method are: (1) the different input data, i.e., we use the 2D neighborhood information (rx,ry) in addition to the 3D points P(X,Y,V) (cf. Figure 5a with Figure 9). In the TMRP method, the triangles are spanned based on the 2D neighborhoods of the 3D points; see Figure 6. (2) Moreover, the interpolation method we present is more accurate.

## 6. Qualitative and Quantitative Experiments

### 6.1. Density and Accuracy

We evaluate our algorithm based on density and accuracy using four different measurement setups and application areas (cf. Figure 3).

#### 6.1.1. Experiment #1 —Qualitative Comparison of Simple SOTA Proj., DT_nea_, and TMRP

In this experiment, we demonstrate the advantages of our TMRP method for fusing Velodyne Light Detection and Ranging (LiDAR) and monocular camera data for depth maps using the established benchmark stereo dataset KITTI-2012 [27]. Figure 10 shows the measurement setup consisting of Velodyne HDL-54E LiDAR and two monocular cameras. For data fusion, the sparse Velodyne point cloud $P_{\mathrm{source}}\left(X,Y,Z\right)$ must be constrained to the corresponding camera section $P_{\mathrm{source}-\mathrm{in}-\mathrm{FOV}}\left(X,Y,Z\right)$. Then, the transformation to the image coordinate system of the target camera ${\mathrm{cam}}_{0}$ is performed. The mathematical details of the transformation [62], as well as the generation of the raster information rx and ry, are described in the Appendix A. The transformed points P(X,Y,Z) resp. P(X,Y,Z,rx,ry) are the input data for the projection methods.

Figure 11 shows the comparison of the simple SOTA projection and our TMRP method. The depth map based on the simple SOTA projection method results in a sparse disparity map. Our TMRP enables the generation of a dense, accurate depth map.

Figure 12 shows the qualitative comparison of another method, the polygon-based method using Delaunay triangulation and nearest-neighbor interpolation (DT_nea_) [7,8]. Although the DT_nea_ method can produce dense maps with almost 100 % density, these maps contain errors. For example, valid gaps, such as car windows, are not considered (see challenge (A), Figure 1). (Transparent objects are a challenge for conventional sensors in the visible spectrum [2]. Transparency-conscious consideration is imperative and a corner case in the stereo image [1,36].) In contrast to DT_nea_, our TMRP never creates false neighbors (C). Moreover, our TMRP keeps all valid gaps (A) caused by the physical limitations of the capturing cameras.

The TMRP method interpolates based on the 2D neighborhood of the source point cloud. For line scanners, this 2D neighborhood is obtained from the spherical coordinates (see Appendix A). Figure 12 *(right)* shows the effect of different horizontal angular resolutions. With different horizontal angular resolutions, the density of the output sourcerasterimage is different (Figure 5a). At 0.09 ${}^{\circ }$, the points are mostly on every second grid point, so there are no neighbors (see Figure 6). At 0.18 ${}^{\circ }$, almost every grid point has a value, so there are neighbors between which interpolation is possible.

#### 6.1.2. Experiment #2—Focus on Challenge (A) Resp. (C) Using a Test Specimen

In some applications, it is very important that gaps in the point cloud caused by physical limitations of the camera technology or by cutouts in the surface are taken into account as completely as possible in the projection method. To minimize challenge (B), we use transformed points based on transformation into equal resolution. Figure 13 shows the results of three projection methods using a special test specimen. The simple SOTA projection gives the worst results. The closed source algorithm of the blaze SDK achieves better results. Our algorithm, on the other hand, performs best.

#### 6.1.3. Experiment #3—Focus on Challenge (B) Using a Test Specimen

Figure 14 shows the results based on the TMRP and the SOTA projection in terms of density and accuracy using a transformed, low-resolution point cloud. In this experiment, the resolution of the target image is higher than the resolution of the source image by a factor of 3.6. To quantitatively compare the methods, we consider in each case both the entire target image and a region of interest (ROI) that includes the captured area of our test specimen, a white matte plane. We show the influence of the individual challenges with difference images (pixel-by-pixel calculation). The SOTA-based target image does not consider 0.9 % of the total pixels as a valid gap—challenge (A) resp. (C). In addition, 24.7% false gaps were generated—challenge (B). This reduces the density of the entire image by a factor of 1.57 compared to the TMRP-based image. If you look at the defined ROI, a section without valid gaps, the density decreases by a factor of 1.61.

#### 6.1.4. Experiment #4—Focus on Influence of Points with Transformation into Equal/Unequal Resolution

Data-driven methods [15,36] require datasets with (dense) ground truth. To create a real (non-synthetic) ground truth, the optical measurement setup consists of at least the actual sensor (target coord. system) and a reference sensor (source). Our TMRP is a suitable method to produce dense, accurate ground truth images. We demonstrate the utility of our method using an image from our *TranSpec3D* (created in cooperation with Fraunhofer IOF and Technische Universität Ilmenau, publication still pending) stereo dataset of transparent and specular objects. Figure 15 shows the measurement setup and input data. Figure 16 shows the results of TMRP and SOTA proj. based on transformed points to equal resolution (*top*) and to higher resolution (*bottom*). In the case of “transformation to equal resolution” (*top*), the target sensor corresponds to the source sensor. Thus, the challenge (B) does not occur. However, 3.9% of the total pixels are not considered as gaps. In the case “transformation to a higher resolution” (bottom), the target sensor does not match the source sensor. In our experiment, the resolution of the target image is higher than the resolution of the source points by a factor of 5.39. Thus, on the one hand, only 1.0% of the total pixels are not considered as gaps (cf. to equal resolution). On the other hand, there are 29.6% false gaps in the target image based on the SOTA projection. This reduces the density of the SOTA-based target image (16.2%) by more than half compared to the TMRP-based target image (34.8%).

### 6.2. Computation Time, Memory Usage and Complexity Class

Our TMRP algorithm is versatile (Section 1.2) and takes into account all four challenges. Currently, the TMRP algorithm is programmed in such a way that the pixels are calculated sequentially. However, the TMRP algorithm is highly parallelizable—all pixels can be calculated in parallel (see Appendix B). Table 1 shows the performance of the SOTA proj. and our TMRP algorithm based on computation time and memory usage.

We determine the complexity of the TMRP algorithm using the 𝒪 notation (upper bound of the complexity of an algorithm). Table A1 shows the 𝒪 notation for the individual Appendix B. Equation (Equation 2) shows the resulting total running time of the TMRP algorithm. In this notation: ${\mathrm{width}}_{\mathrm{src}/\mathrm{target}}$ and ${\mathrm{height}}_{\mathrm{src}/\mathrm{target}}$ stand for the width and height of the source_raster_image/target_image; *n* stands for the number of points in point_list in Appendix B; *n* stands for the size of the vector_image and *m* stands for the size of the raw_pixel_list in Appendix B; and *n* stands for the number of elements in raw_pixel_list in Appendix B. The 𝒪 notation in Equation (Equation 2) as well as the direct measurement of the runtime [64] shows that the algorithm has a linear time complexity in relation to the source and target image resolution (in px).
(2)$$𝒪(\underset{A1}{\underbrace{n^{2}}}+\underset{A2}{\underbrace{{\mathrm{width}}_{\mathrm{src}}\;\cdot \;{\mathrm{height}}_{\mathrm{src}}}}+\underset{A9}{\underbrace{n\;\cdot \;m}}+\underset{A10}{\underbrace{{\mathrm{width}}_{\mathrm{target}}\;\cdot \;{\mathrm{height}}_{\mathrm{target}}\;\cdot \;n}})$$

## 7. Conclusions, Limitations, and Future Work

### 7.1. Conclusions

Our presented Triangle-Mesh-Rasterization-Projection (TMRP) algorithm allows the projection of transformed points onto a dense, accurate 2D raster image. To achieve dense accuracy, points with original raster information P(X,Y,V,rx,ry) are required. The original 2D neighborhood information (rx,ry) can be acquired simultaneously with almost all 3D measurement methods. Since the 2D neighborhood of the 3D points are known, the triangular interpolation can be performed quickly. Valid gaps that exist in the original 3D survey due to the physical limitations of the camera technology used (Figure 1, (A)) are fully considered. Furthermore, false gaps (B) or false neighborhoods (C) are never generated. However, there is a physical limitation to challenge (C), the generation of false neighbors. If the Nyquist–Shannon sampling theorem is not observed, then of course there will be false neighbors. Additionally, ambiguities (D) are taken into account, and foreground or background are clearly separated depending on the application (Figure 7). This method can also be used to build a high-quality 3D model (see Figure 4). Table 2 shows the advantages and disadvantages of our TMRP compared to state-of-the-art single-shot projection methods. In contrast to common single-shot methods (Section 2), the TMRP algorithm solves challenges (A–D), and the density is independent of XZY. However, the 2D neighborhood raster information of the source points (rx,ry) is necessary here. Our software TMRP is available at http://github.com/QBV-tu-ilmenau/Triangle-Mesh-Rasterization-Projection (accessed on 1 July 2023). Compared to cloud-source methods from sensor manufacturers, the TMRP method is applicable independently of the sensor and modality. Moreover, it is freely available and open source (To verify the performance of the TMRP method, we compared it with the closed source method of Microsoft’s Azure Kinect RGB-D sensor. “The qualitative comparison shows that both methods produce almost identical results. Minimal differences at the edges indicate that our TMRP interpolation is more accurate.” [64]). This aspect of the different modalities is becoming increasingly important, for example, in the detection of optically uncooperative objects in the visual spectral range [2,42] or in trusted human–robot interactions [13,14].

We believe that our TMRP method will be highly useful for various image analysis and enhancement tasks (Figure 3). To show the potential of the TMRP algorithm, we applied it to input data for various use cases (see Section 6): (1) Data fusion of sparse LiDAR points and monocular camera data using the established KITTI-2012 dataset [27]. (2) Comparison to a closed source SDK function of a low-cost depth sensor. (3) Applications to create a real (stereo) training dataset for deep neural networks using our new *TranSpec3D* dataset of transparent and specular objects without object painting (cf. [36,40,41]).

### 7.2. Limitations for Online Applications

Currently, TMRP is implemented as a sequential algorithm (Table 1). However, the TMRP is highly parallelizable, which will result in shorter calculation times in the future.

### 7.3. Future Work

The algorithm has the potential—due to overcoming the limitations of existing single-shot projection methods, the independence from the sensor manufacturer, as well as the high parallelizability—to increase the processing stability of various image analysis and enhancement tasks (Section 1.2). In the future, however, the benefit of TMRP must be proven. On the one hand, we want to show the utility for robotics applications in terms of grasp planning and object recognition using different low-cost RDB-D sensors. On the other hand, we want to show the benefit of TMRP for deep learning approaches. For this use case, in the long term, our TMRP algorithm should help to provide more real datasets without object painting that fully represent reality, thus reducing the differences seen with synthetic data.

## Figures and Tables

**Figure 1 sensors-23-07030-f001:**
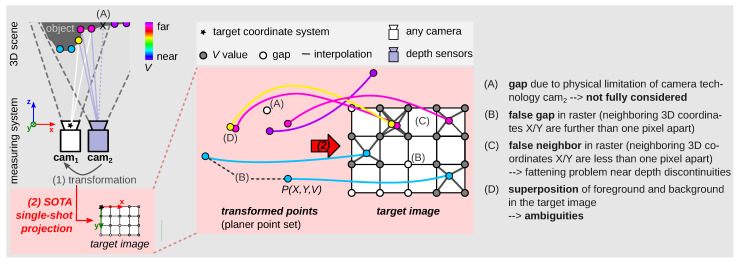
Challenges in common single-shot projection methods. Projecting points P(X,Y,V) onto a 2D raster image, called *target image*. Measurement setup, e.g., consisting of a first camera cam_1_ (with target coordinate system) and a second sensor cam_2_ (source: low resolution point cloud). (1) The captured raw points of cam_2_ are transformed into the target coord. system. (2) These transformed points are projected onto a 2D raster image (*target image*) using state-of-the-art (SOTA) projection methods (Section 2). The target image is not densely accurate due to errors (**A**,**B**) [3], (**C**), [3,4,5,6] and (**D**) [7]. Challenge (**D**) is due to perspective view of the technology, e.g., LiDAR, [7] or (ii) due to different viewing angles of the two cameras resp. sensors (cam_1_≠ cam_2_). Depending on the hardware used (equal/unequal), some challenges occur more strongly/weakly or not at all.

**Figure 2 sensors-23-07030-f002:**
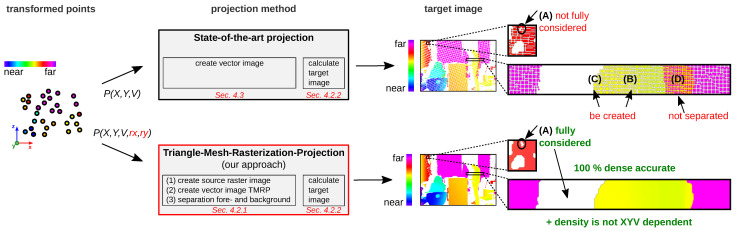
Comparison of state-of-the-art (SOTA) projection (top) and our Triangle-Mesh-Rasterization-Projection (TMRP) method (bottom). For the SOTA projection method, the transformed points P(X,Y,V) are used as input (left). TMRP requires transformed points with an additional property of the points (left): the raw 2D raster information of the points P(X,Y,V,rx,ry). rx and ry can be used to perform a dense, accurate interpolation between the 2D pixels that were adjacent in 3D. Both generated target images are shown on the right. Challenges (**A**–**D**) are described in Figure 1. In the target image (bottom-right), there are only valid gaps (**A**), which are also present in the original measurement of the 3D data. Input data: transformed point cloud with 54.9 % valid points; Time-of-Flight (ToF) sensor (DepthSense™ IMX556; ≈0.3 Mpx); target image resolution: 1280 px × 864 px.

**Figure 3 sensors-23-07030-f003:**
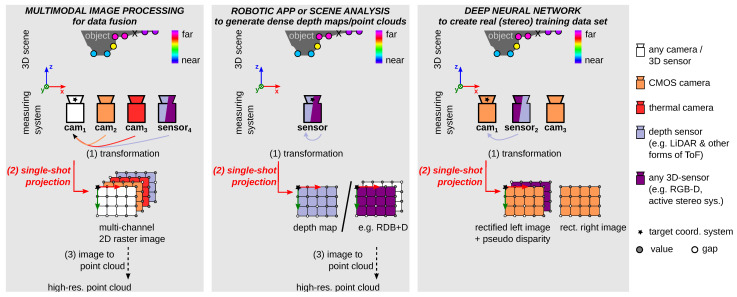
Three generalized measurement systems for different use cases (Section 1.2) of single-shot projection methods in which the challenges occur (Figure 1). In general, for multimodal image processing (left): for data fusion/registration. For robotic applications or scene analysis (mid): to generate dense depth maps or point clouds when using a low-resolution depth sensor, e.g., Basler blaze-101. For deep neural networks (right): to create a real (stereo) training dataset with ground truth disparity maps.

**Figure 4 sensors-23-07030-f004:**
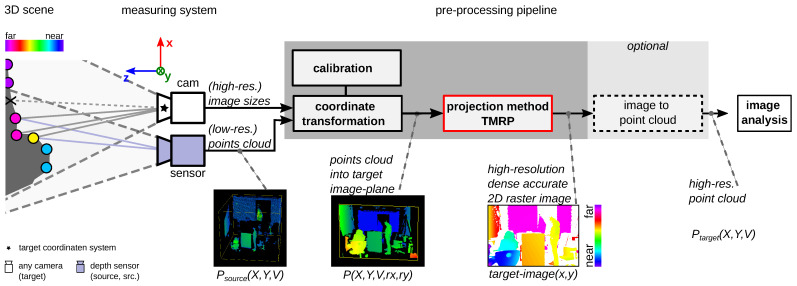
Embedding the TMRP method in the process pipeline. Data acquisition from raw points (source sensor); (1) coordinate transformation into target coordinate system (cam); (2) project transformed point cloud P(X,Y,V,rx,ry) onto a dense, accurate target image via TMRP; (3) create a high-quality 3D model (optional): target image to point cloud $P_{\mathrm{target}}\left(X,Y,Z\right)$.

**Figure 5 sensors-23-07030-f005:**
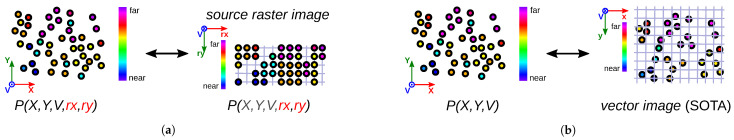
Create sourcerasterimage (using TMRP) and vector_image (using SOTA projection). (**a**) Part of TMRP: Create sourcerasterimage based on transformed points P(X,Y,V,rx,ry) (Appendix B). (**b**) Part of SOTA proj.: Create vector_image based on transformed points P(X,Y,V) (Appendix B).

**Figure 6 sensors-23-07030-f006:**
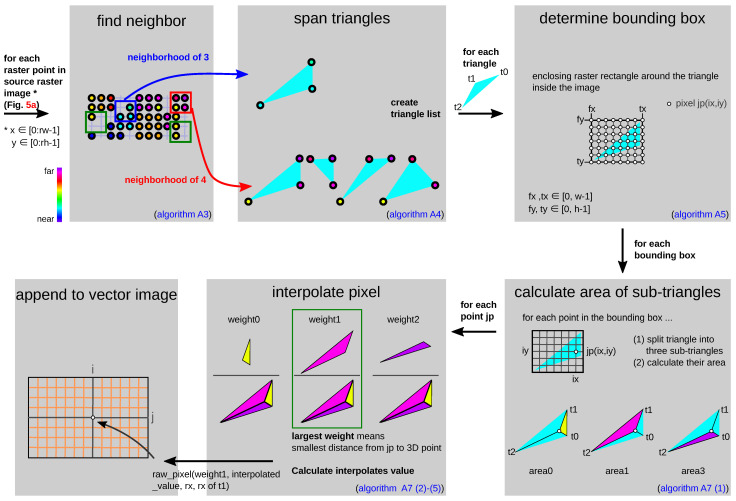
Pipeline to create vector_image (using TMRP). Find 3D point P(X,Y,V,rx,ry) with smallest distance to grid point jp. Appendix B describes this pipeline mathematically.

**Figure 7 sensors-23-07030-f007:**
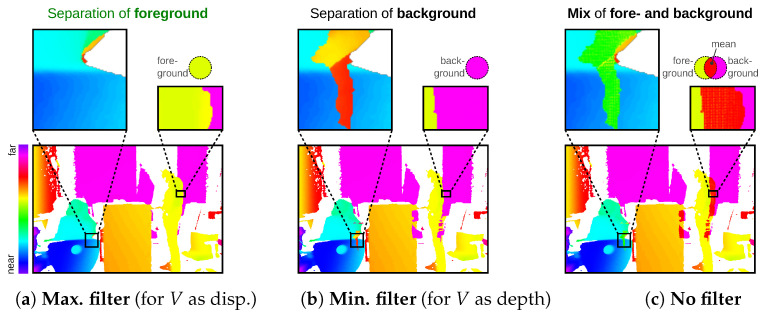
Separation of foreground and background—to solve challenge (D), see Figure 1. The choice of filter (**a**–**c**) depends on value *V* of the points. Used input points with the *V* as disparity in (**a**–**c**).

**Figure 8 sensors-23-07030-f008:**
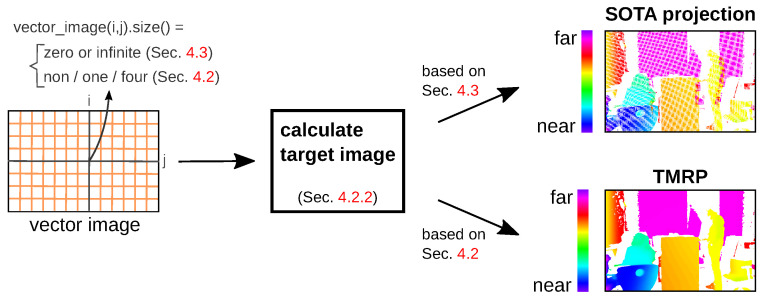
Create target_image (right) based on input vector_image (left) calculated from the SOTA projection (Section 4.3) or TMRP (Section 4.2.1).

**Figure 9 sensors-23-07030-f009:**
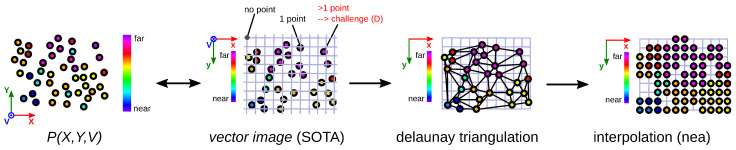
Schematic representation of the polygon-based method using Delaunay triangles and nearest neighbor (DT_nea_), described in [7]. To increase the computing power, the points (finite real numbers) can be rounded to integer values in the vector_image.

**Figure 10 sensors-23-07030-f010:**
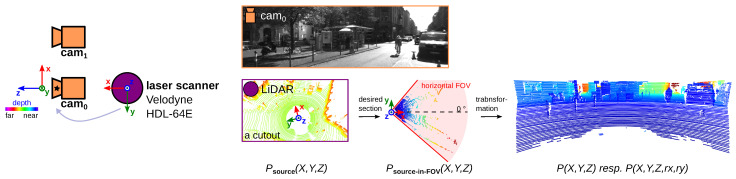
Measurement setup of experiment #1. (left) The established KITTI-2012 dataset [27] was generated using a Velodyne LiDAR and a passive stereo camera system (cam_0_ and cam_1_; 2× Point Gray FL2-14S3M-C), among others. (right) For the data fusion of the low-resolution LiDAR point cloud $P_{\mathrm{source}}\left(X,Y,Z\right)$ into the left camera cam_0_, the point cloud must be limited to the desired section $P_{\mathrm{source}-\mathrm{in}-\mathrm{FOV}}\left(X,Y,Z\right)$. This is achieved through a vertical FOV [ −24.9${}^{\circ }$, 2.0${}^{\circ }$] and a horizontal FOV [ −45${}^{\circ }$, 45${}^{\circ }$]. Afterwards, these points have to be transformed into the coordinate system of cam_0_. Input data: Frame 89, KITTI 2011_09_26_drive_0005 [27]. Raster information rx based on 0.18${}^{\circ }$ horizontal angular resolution. Target img. res: 1242 px ×375 px.

**Figure 11 sensors-23-07030-f011:**
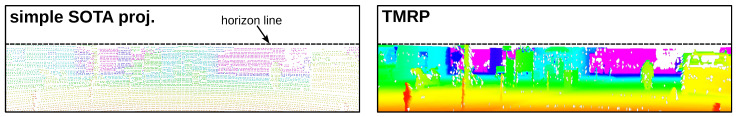
Qualitative comparison of depth maps based on simple SOTA proj. (left) and TMRP (right) method using KITTI-2012 dataset. The drawn horizon line results from the vertical FOV [ −24.9${}^{\circ }$, 2.0${}^{\circ }$]. Input data: see Figure 10.

**Figure 12 sensors-23-07030-f012:**
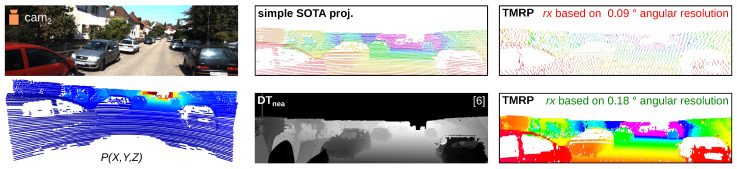
Qualitative comparison of different projection methods. (left) RGB image (${\mathrm{cam}}_{3}$) and corresponding transformed points P(X,Y,Z) (Velodyne HDL-64E, with vertical FOV [ −24.9${}^{\circ }$, 2.0${}^{\circ }$] and horizontal FOV [ −45${}^{\circ }$, 45${}^{\circ }$]) of frame 20, KITTI 2011_09_26_drive_0064 [27] (cf. Figure 10). (mid-right) Different projection methods: (top) simple SOTA projection; (bottom) polygon-based method using Delaunay triangulation and nearest-neighbor interpolation (DT_nea_) [8]. (right) Our TMRP method based on 0.09${}^{\circ }$ and 0.18${}^{\circ }$ horizontal angular resolution. For better visualization, the focus of the false color display was placed on the closer depth values.

**Figure 13 sensors-23-07030-f013:**
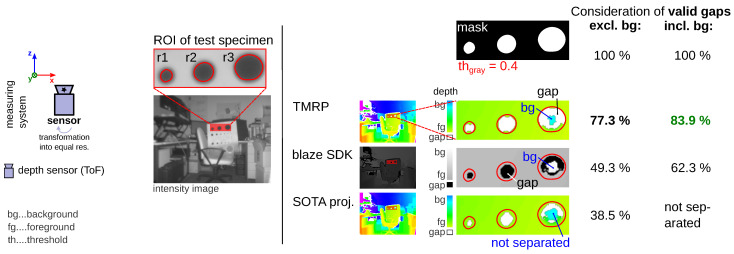
Comparison of TMRP, closed source algorithms of blaze SDK (v4.2.0), and SOTA proj. regarding challenge (A) resp. (C) (see Figure 1). (left) Measuring setup. (mid) ToF intensity image with region of interest (ROI) of the test specimen. (right, top) Binary mask and (right, bottom) depth maps. Test specimen: White matte plane with cutouts (r1/2/3=3/4/5mm; made with a laser cutter). Input data: Transformed points: 91.0 % valid points of ≈0.3 Mpx ToF camera (Basler blaze-101 ToF sensor with DepthSense™ IMX556PLR sensor); target img. res.: 640 px × 480 px. Consideration of: valid gaps, incl. background (bg). Result: Our TMRP algorithm considers 83.9 % of cutouts (r1/2/3). The closed source blaze SDK algorithm also separates the bg correctly. TMRP considers 21.6 % more valid pixels (in relation to cutouts) compared to the blaze SDK algorithm. Why not 100%? Depends significantly on the entered point cloud and on the selected threshold (binary mask).

**Figure 14 sensors-23-07030-f014:**
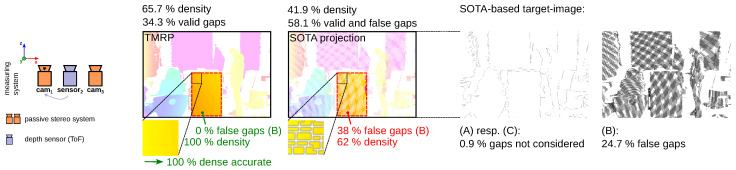
Comparison of TMRP and SOTA proj. regarding challenge (B) (see Figure 1). Test specimen: White matte plane. Input data: Transformed points: 54.9% valid points of ≈0.3 Mpx ToF sensor (DepthSense™ IMX556); target img. res.: 1280 px ×864 px. TMRP: 100 % dense, accurate due to the use of the raw 2D raster information of the input points never creates false gaps/neighbors in raster—Appendix B).

**Figure 15 sensors-23-07030-f015:**
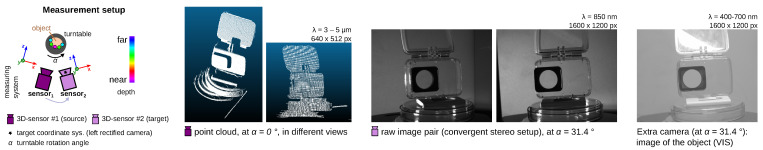
Measurement setup of experiment #4: (left) We use our novel measurement principle ${}^{†}$ to create a real transparent and specular stereo dataset without SOTA object painting (cf. [36]). sensor_1_: 3D thermal infrared sensor technology of Landmann et al. [42] (a.o. 2 × FLIR A6753sc). sensor_2_: 3D sensor [63] (a.o. 2 × Blackfly® S USB3). Input data: (mid) Sample #0157 of TranSpec3D dataset ${}^{†}$; sensor_1_: point cloud with ≈0.2 Mpx; source img. res.: 696 px ×534 px. sensor_2_: target img. res.: 1616 px ×1240 px (≈2.0 Mpx). Object: (mid, top-down) Transparent waterproof case for action camera, Petri dish (glass), and polymethyl methacrylate (PMMA) discs with different radii. ${}^{†}$ Stereo dataset (real, laboratory) created in cooperation with Fraunhofer IOF and Technische Universität Ilmenau (publication still pending).

**Figure 16 sensors-23-07030-f016:**
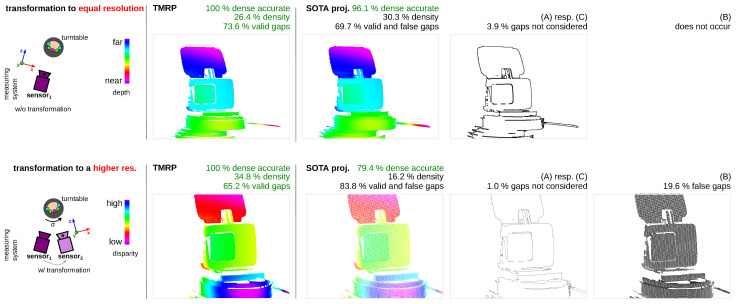
Results of projection methods based on transformed points with equal (top) and higher resolution (bottom). Input data: TranSpec3D dataset^†^. Result: (top) In the target image based on the SOTA proj., 3.9% of the total pixels are not considered gaps. (bottom) In the target image based on the SOTA proj., 1.0% of the total pixels are not considered gaps. Due to the different sensor resolutions of sensor_1_ and sensor_2_, there are 29.6% false gaps in the target image based on the SOTA projection. This reduces the density of the SOTA-based target image (16.2%) by more than half compared to the TMRP-based target image (34.8%). ${}^{†}$ Stereo dataset (real, laboratory) created in cooperation with Fraunhofer IOF and Technische Universität Ilmenau (publication still pending).

**Table 1 sensors-23-07030-t001:** Quantitative comparison of computation time and memory usage. Avg. computation time and max. resident set size (RSS) on processing unit (i9): Intel Core i9-7960X CPU @ 2.80 GHz and (i7): Intel Core i7-6700X CPU @ 4.00 GHz. Input data: transformed points (see Figure 2): 54.9 % valid points of ≈0.3 Mpx ToF sensor (Basler blaze-101); target img. res.: 1280 px ×864 px (≈1.1 Mpx).

Algorithm (Sequential)	Computation Time		Max. RSS	Density	Dense, Accurate
	Unit (i9)	Unit (i7)	Unit (i9)/(i7)	("visual")	
SOTA proj.	**0.183 s**	**0.254 s**	**58.2 MiB**	41.9 %	63.8 %
TMRP	0.454 s	0.405 s	106.8 MiB	**65.7 %** ${}^{†}$	**100 %**

${}^{†}$ Why not 100 %? This value depends on the density of the input data. The remaining 34.3 % are valid gaps (A).

**Table 2 sensors-23-07030-t002:** Qualitative comparison of single-shot projection methods (Section 2). (left-to-right) State-of-the-art projection (SOTA proj., Section 4), Joint bilateral upsampling (JBU) [6], Closed source blaze SDK projection (v4.2.0), Polygon-based method using Delaunay triangles and nearest neighbor ($DT_{nea}$) [7], and our TMRP algorithm (Section 4).

Properties	SOTA Proj.	JBU [6]	Blaze SDK	DT_nea_ [7]	TMRP (Ours)
Creates false neighbors in raster; Figure 2A,C	low	often	less	often	**never ${}^{§}$**
Creates false gaps in raster; Figure 2B	often	low	low	**never**	**never**
Resolution of ambiguities; Figure 2D	no	no	**yes**	**yes**	**yes**
Density is independent of XYZ	no	no	closed source	**yes** ${}^{*}$	**yes** ${}^{*}$
Required input	**w/o rx,ry**	**w/o rx,ry**	closed source	**w/o rx,ry**	w/ rx,ry
Computing effort	**low**	middle	middle–high ${}^{‡}$	high	high ${}^{†}$
For almost any sensor & modality	**yes**	**yes**	no	**yes**	**yes**
Code available (open source & free)	**yes**	**yes**	no	**yes**	**yes**

${}^{§}$ TMRP method based on interpolation considering the source 2D neighborhood (see Section 4.2.1) → physical limitation: If the Nyquist–Shannon sampling theorem is not observed, i.e., the valid gaps are below the Nyquist frequency, then of course false neighbors are present. ${}^{†}$ However, highly parallelizable. ${}^{‡}$ Image processing is performed on a very powerful NXP processor on the ToF camera. ${}^{*}$ This method interpolates all points regardless of the distance between the points of a triangle.

## Data Availability

Our software TMRP with sample data is available at http://github.com/QBV-tu-ilmenau/Triangle-Mesh-Rasterization-Projection (accessed on 1 July 2023).

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
