# Peer review of "Triangle-Mesh-Rasterization-Projection (TMRP): An Algorithm to Project a Point Cloud onto a Consistent, Dense and Accurate 2D Raster Image"

_sensors, 2023, doi:10.3390/s23167030_

Round 1

Reviewer 1 Report

The authors of the article propose an interesting solution in the field of combining information from several different sensors - the so-called Data fusion problem. In truth, however, they prefer to describe the problem with the applications in which it applies and with the specific implementation - as an algorithm to project a point cloud into a consistent and dense accurate 2D raster image. The article describes the problem perfectly. The authors provide a number of very illustrative figures that contribute to a good understanding of their concept. Moreover, they implement pseudocode of the algorithms and cite a publicly available repository of their code with which readers will be able to replicate the experiments. This brings the article closer to my ideal of one where any reader could replicate the results obtained and be convinced of their effectiveness.

Unfortunately, the article also has some flaws, the removal of which will further improve the qualities of the article.

1. The article has the structure and organization of a presentation in which beautifully designed slides are commented on. In a classically designed article, the text is leading and the figures are an illustration to it.

2. The information provided by the authors is somewhat chaotically organized. In 1. Introduction, in addition to a description of the solved problem, a comparison with the classical solution (the so-called state-of-the-art (SOTA) projection) is also given. This is followed by a description of the many application areas and a definition of the main achievements in the article. In 2. Related work again follows a comparison with SOTA and other approaches.

1. Some repetitions should be avoided such as: Our TMRP completely solves the SOTA challenges (A) to (D).

2. The pronunciation in some sentences should be improved and syntactical errors such as "the vetor image" should be corrected.

Author Response

Dear Reviewer,

Thank you for the helpful Comments and Suggestions.

Comments and Suggestions for Authors

  • Re 1: Yes, that's true - the illustrations are dominant, especially in the back half of the documentation. However, I did not want to produce more text, as the paper is already very extensive.
  • Re 2: I have adjusted the introduction a little so that it is easier to see what is to follow.

Comments on the Quality of English Language:

  • Re 1.: Yes, I have now revised that.
  • Re 2: I have looked through the whole publication again and corrected the typing errors. 

Thank you very much!

Yours sincerely,
Christina Junger

Reviewer 2 Report

The paper presents a new method called Triangle-Mesh-Rasterization-Projection for projecting 3D point clouds into accurate, dense 2D images. This has applications in multimodal image processing, robotics, scene analysis, and generating training data.

The proposed technique uses the 3D point coordinates along with 2D neighborhood information to perform fast triangulation interpolation. This allows it to overcome limitations of existing single-shot projection methods:

- It eliminates false gaps or incorrect neighborhoods that those methods can produce.

- The density of the projected image is independent of XYZ coordinates.

- It resolves ambiguities present in other techniques.

The authors claim their method is able to solve all current challenges related to point cloud projection, generating high quality 2D images without valid gaps.

Use cases demonstrating the usefulness of this technique for recognition tasks and manufacturing process optimization are mentioned but not detailed in the abstract.

Overall, the key novelty presented is the Triangle-Mesh-Rasterization-Projection approach for improved point cloud to 2D projection, providing both accuracy and density. The authors aim to advance image analysis and robotic applications through this contribution. This paper is well motivated and well written. I have the following minor comments. 

The statement that this method can "solve all current challenges" is quite bold and needs qualification or tempering. Clarifying the specific limitations or edge cases where gaps/ambiguities may still occur would add more nuance.

The use cases demonstrating usefulness are mentioned but not described in any detail. Expanding on 1-2 examples showing where this technique improves recognition tasks or manufacturing processes would make the impact more concrete.

The writing could be tightened in places - for example, "highly relevant" and "well known" are vague phrases. More direct and specific language would strengthen the abstract.

The authors may want to highlight 1-2 of the most novel aspects of their technique compared to prior work in the final sentence of the abstract to leave reviewers with a clear sense of the key contributions.

The English is OK!

Author Response

Dear Reviewer,

Thank you for the helpful Comments and Suggestions.
I have taken all comments into account and incorporated them.

Comments and Suggestions for Authors

  • I have modified the abstract and have included all comments. This is a very good idea - thank you very much.
  • I have softened the statement "solve all current challenges". I have also specified a boundary condition. 
  • I have replaced the expression "highly relevant" and "well known".

Thank you very much!

Yours sincerely,
Christina Junger